# Isoniazid preventive therapy during infancy does not adversely affect growth among HIV-exposed uninfected children: Secondary analysis of data from a randomized controlled trial

**Ashenafi Shumey Cherkos**[1], **Sylvia M. LaCourse**[2,3,4], **Daniel A. Enquobahrie**[2], **Jaclyn N. Escudero**[4], **Jerphason Mecha**[5], **Daniel Matemo**[6], **John Kinuthia**[6,7], **Sarah J. Iribarren**[8], **Grace John-Stewart**[2,3,4,9] *

1 Department of Population and Community Health, School of Public Health, University of North Texas Health Science Center, Fort Worth, Texas, United States of America, 2 Department of Epidemiology, School of Public Health, University of Washington, Seattle, Washington, United States of America, 3 Division of Allergy and Infectious Diseases, Department of Medicine, University of Washington, Seattle, Washington, United States of America, 4 Department of Global Health, School of Public Health, University of Washington, Seattle, Washington, United States of America, 5 Centre for Respiratory Diseases Research, Kenya Medical Research Institute, Nairobi, Kenya, 6 Medical Research Department, Kenyatta National Hospital, Nairobi, Kenya, 7 Department of Obstetrics and Gynaecology, Kenyatta National Hospital, Nairobi, Kenya, 8 Biobehavioral Nursing and Health Informatics, School of Nursing, University of Washington, Seattle, WA, United States of America, 9 Department of Pediatrics, University of Washington, Seattle, Washington, United States of America

* gjohn@uw.edu

## Abstract

### Background

Isoniazid preventive therapy (IPT) decreases risk of tuberculosis (TB) disease; impact on long-term infant growth is unknown. In a recent randomized trial (RCT), we assessed IPT effects on infant growth without known TB exposure.

### Methods

The infant TB Infection Prevention Study (iTIPS) trial was a non-blinded RCT among HIV-exposed uninfected (HEU) infants in Kenya. Inclusion criteria included age 6–10 weeks, birthweight ≥2.5 kg, and gestation ≥37 weeks. Infants in the IPT arm received 10 mg/kg iso-niazid daily for 12 months, while the control trial received no intervention; post-trial observa-tional follow-up continued through 24 months of age. We used intent-to-treat linear mixed-effects models to compare growth rates (weight-for-age z-score [WAZ] and height-for-age z-score [HAZ]) between trial arms.

### Results

Among 298 infants, 150 were randomized to IPT, 47.6% were females, median birthweight was 3.4 kg (interquartile range [IQR] 3.0–3.7), and 98.3% were breastfed. During the 12-

**Data Availability Statement:** Data cannot be shared publicly because of ethical restrictions in

the informed consent documents and in the approved human subject's protection plan of the University of Washington and the University of Nairobi/Kenyatta National Hospital Ethics and Research Committee. Data are available from the Global Center for Integrated Health of Women, Adolescents, and Children (Global WACh) Institutional Data Access / Ethics Committee for researchers who meet the criteria for access to confidential data. Interested individuals should contact: STEPHANIE EDLUND-CHO, MSW Program Operations Specialist | Global WACh Department of Global Health Hans Rosling Center Box 351620 3980 15th Ave NE, Seattle, WA 98195 office: 206.685.6809 globalwach.org.

**Funding:** This work was supported by the Thrasher Research Fund, National Institute of Allergy and Infectious Diseases (NIAID), Fulbright program awarded to the Northern Pacific Global Health Fellows Program by the Fogarty International Center of the National Institutes of Health (NIH/ Fogarty), and National Center for Advancing Translational Sciences at National Institutes of Health (NIH) (Thrasher to GJ-S, NIH/NIAID K23AI120793 to SML, NIH/NIAID 2K24AI137310 to TRH, NIH/Fogarty R25TW009345 to AJW and NIH UL1TR000423 for REDCap). ASC had a diversity supplement grant from NIH (NIH/NIAID 1R01AI142647).

**Competing interests:** The authors have declared that no competing interests exist.

month intervention period and 12-month post-RCT follow-up, WAZ and HAZ declined significantly in all children, with more HAZ decline in male infants. There were no growth differences between trial arms, including in sex-stratified analyses. In longitudinal linear analysis, mean WAZ ($\beta$ = 0.04 [95% CI:-0.14, 0.22]), HAZ ($\beta$ = 0.14 [95% CI:-0.06, 0.34]), and WHZ [$\beta$ = -0.07 [95% CI:-0.26, 0.11]) z-scores were similar between arms as were WAZ and HAZ growth trajectories. Infants randomized to IPT had higher monthly WHZ increase ($\beta$ to 24 months 0.02 [95% CI:0.01, 0.04]) than the no-IPT arm.

## Conclusion

IPT administered to HEU infants did not significantly impact growth outcomes in the first two years of life.

## Introduction

Infection with *Mycobacterium tuberculosis* (Mtb) before the age of 2 years can progress to severe TB disease, with rapid progression occurring within the first year of Mtb infection [1–5]. Without treatment, children with latent Mtb infection, also called latent TB infection (LTBI), have about a 19% risk of developing TB within two years [1–4, 6, 7]. HIV-exposed uninfected (HEU) children have a higher risk of TB and Mtb infection than HIV-unexposed uninfected (HUU) children [8]. Africa accounted for 30.5% of the global pediatric TB burden in 2018 [9] and 37.0% of TB-related deaths among children under 15 years of age. Therefore, approaches to prevent and treat TB infection in TB endemic areas among HEU children are a priority.

Isoniazid preventive treatment (IPT) decreases the progression of Mtb infection to TB disease [10–12]. The World Health Organization (WHO) recommends IPT in children under 5 years with known TB exposure and children with HIV (CHIV) older than one year of age [13]. Recent studies suggest that household exposure accounts for less than 30% of Mtb transmission to children, with most occurring without known contact [14]. This suggests that WHO recommendations may miss many children at risk for TB. Because progression from Mtb infection to TB disease is rapid during infancy, IPT administration, regardless of contact history, may be warranted in some higher-risk groups of infants. We conducted a randomized trial (RCT) (NCT02613169) to examine the impact of IPT on the incidence of Mtb infection in HEU infants [15]. In the primary trial, IPT was associated with a non-significant trend for decreased Mtb infection (predominantly assessed by tuberculin skin tests (TST)) during the first year of life [16], which was not sustained up to 24 months of age [17].

While IPT has clear benefits in decreasing the risk of TB disease, it has side effects that can potentially compromise caloric intake, including loss of appetite, nausea, vomiting, and upset stomach [18]. These, in turn, could influence infant growth. Although trials of IPT in children have shown no signs of an increased risk of adverse events [17], the effects of IPT on the growth of infants remain unclear. In observational studies, assessing the impact of IPT on infant growth has been challenging due to confounding by indication. IPT is prescribed to children who have been exposed to TB or HIV-infected children regardless of their exposure. Consequently, children receiving IPT may have different characteristics that influence their growth than those who do not receive IPT.

The infant TB Infection Prevention Study (iTIPS) provided a unique opportunity to compare growth in HEU infants receiving IPT versus those not receiving IPT in a randomized

controlled trial, unconfounded by known TB exposure or other related characteristics. There-fore, we leveraged the RCT design of the parent study to rigorously examine the effect of IPT on infants' growth over the first two years of life.

## Materials and methods

### Parent trial design and intervention

This secondary analysis used data from the iTIPS trial, a non-blinded RCT among HIV-exposed infants in Western Kenya. As reported in detail previously [15], randomization was stratified by site and generated by the study statistician using a computer-generated random block size. Inclusion criteria for the parent study were 6–10 weeks of age, birth weight $> = 2.5$ kg, and not premature ($>37$ weeks of gestation). Exclusion criteria included known household TB exposure, including mothers with a history of TB in the past year and infants enrolled in other TB prevention programs or vaccine studies. Infants 6–10 weeks of age in the IPT arm initiated isoniazid 10 mg/kg once daily for 12 months. The control arm received no interven-tion. Standardized weight-based isoniazid dosing (by weight band using 100 mg scored tablets) was used, corresponding to Kenya and WHO recommendations [15]. Pyridoxine was pro-vided to children randomized to isoniazid to decrease peripheral neuropathy risk. To ensure full dose usage and ease of infant administration, caregivers were advised to pulverize isoniazid and pyridoxine and mix them with small quantities of breastmilk, clean water, or liquid cotri-moxazole. The Intervention ended after 12 months post-randomization (~14 months of age); post-trial observational follow-up continued until 24 months of age for both arms.

### Participants and study period

Overall, 300 HIV-exposed infants were enrolled in prevention of mother-to-child HIV Trans-mission (PMTCT) clinics in Western Kenya. All institutions in the study were public maternal and child health (MCH) clinics embedded in Kenya's Ministry of Health (MOH). Follow-up visits were conducted at 10 and 14 weeks of age and 6, 9, 12, and ~14 (12 months post-ran-domization) months of age. Observational follow-up continued through 24 months of age. The trial lasted from August 2016 to July 2019, but the observational follow-up continued until September 2020. Enrolled infants with at least follow-up after enrollment were eligible for this secondary analysis. The parent trial and this analysis excluded two infants diagnosed with HIV. We had 298 HEU children for this analysis.

### Ethical approvals

The University of Nairobi/Kenyatta National Hospital Ethics and Research Committee (P571/08/2015), Jaramogi Oginga Odinga Teaching and Referral Hospital Ethical Review Committee, the University of Washington Institutional Review Board (STUDY00000761), and Kenya Pharmacy and Poisons Board approved the study. Caregivers gave informed consent. Caretak-ers gave written informed consent and an external and independent Data and Safety Monitor-ing Board monitored adverse events in the parent trial.

### Infant growth characterization

Infants were enrolled at 6–10 weeks of age, and their birth weight was based on a review of maternal child health cards and maternal reports. Weight and length of children were mea-sured at 6, 10, 14 weeks, 6, 12, ~14 (12 months post enrolment), and 24 months of age. A CDC-Kenya team trained study staff for two days on growth monitoring prior to trial

initiation. Infants were measured twice at each visit, and the average weight and length were rounded to the nearest 0.1 kg and 0.1 cm, respectively.

Weight-for-age z-score [WAZ], weight-for-length z-score [WLZ], and length-for-age z-score [LAZ]) were defined using WHO child growth standards [19]. Growth faltering was defined as underweight WAZ <-2, wasting WLZ <-2, and stunting LAZ <-2 [19].

## Maternal and infant information

Baseline maternal and infant characteristics were collected using standardized questionnaires and medical record abstraction administered by trained study staff. Maternal characteristics, including age in years, educational achievement of mothers, current marital status, employment status, household income, mode of delivery, smoking, alcohol, and drug use during pregnancy, timing of the mother's HIV diagnosis, use of antiretroviral therapy (ART), TB diagnosis and, any maternal IPT used during pregnancy were collected. The following baseline infant characteristics were collected: age in weeks, sex assigned at birth, current breastfeeding status, presence of BCG scar, birth weight, current cotrimoxazole use, and ART use for PMTCT of HIV.

## Statistical analyses

Means and standard deviations (SD) were used to describe normally distributed continuous variables, medians and interquartile ranges (IQR) were used to describe skewed distributions, and frequency and percentage to describe categorical variables. We compared baseline maternal and infant characteristics between IPT and no-IPT arms using a two-sided t-test (the Mann-Whitney U test if assumptions were not met) for continuous variables and the Pearson $\chi^2$ test (Fisher's exact test if assumptions were not met) for categorical variables.

The primary analysis was an intent-to-treat (ITT) analysis, comparing infants' growth (WAZ, HAZ, and WHZ) in IPT and no-IPT arms. Linear regression was used to compare the WAZ, HAZ, and WHZ of children measured at ~14 months of age (12 months post-randomization–the end of the trial) and 24 months of age (extended observational follow-up) between the arms. We also fitted linear mixed-effects models (LMEMs) with autoregression correlation structure, random intercept for subjects, and random slope for follow-up time to compare growth (WAZ, WHZ, HAZ) trajectories between IPT vs no-IPT arms from enrollment to ~14 and 24 months of age.

Linear regression was used to determine factors associated with WAZ, HAZ, and WHZ of children measured at ~14 and 24 months of age, and LMEMs to determine factors associated with WAZ, HAZ, and WHZ of children from enrollment to ~14 and 24 months of age.

## Approach to data missingness

There were missing data in this study: overall WAZ (9.2%), HAZ (9.7%), and WHZ (9.7%). Multiple imputations by chained equations (MICE) [20], which ran a series of regression models for each missing variable conditional upon other specified variables, were used to manage missing data. Baseline infant and maternal characteristics (infant's age, sex, BCG scar, current cotrimoxazole use, baseline WAZ, HAZ, WHZ, and maternal age, education, employment, and marital status) were used in the MICE analyses. We imputed the data 25 times. Pooled parameter estimates and their standard errors were calculated according to Rubin's rules to account for the between- and within-imputation variance [20]. We employed one single imputation model to obtain imputed values for outcomes (WAZ, HAZ, and WHZ) missed at each visit during follow-up. In the imputation models, we specified the appropriate distributions

for each of the variables in the model. In the presence of missing data, the multiple imputation approach should yield unbiased estimates assuming data are missing at random (MAR) [20].

## Results

### Maternal and infant baseline characteristics

Among 298 HEU infants enrolled in the parent trial, 150 were randomized to IPT and 148 to the control no-IPT arm, and all were included in this secondary analysis. Baseline maternal and infant characteristics were previously reported in the primary RCT publication and were uniformly distributed between the randomization arms (Table 1) [16, 17]. The mean (SD) age of mothers was 28 (±5) years, and the median (IQR) monthly income was 10,000 KSH (6K – 15K). Most (86.6% [258/298]) were currently married, 53.3% (159/298) completed primary school or less, and 49.7% (148/298) were unemployed. Few (14.4% [43/298]) mothers

**Table 1. Baseline maternal and infant demographics and clinical characteristics.**

| Characteristics | IPT* (n = 150) | No-IPT*(n = 148) | P-value |
|---|---|---|---|
| **Maternal characteristics** | | | |
| Mean age (SD)–years | 27.6 (5.1) | 28.1 (4.9) | 0.344 |
| Primary school completed or less–no. (%) | 75 (50.0) | 84 (56.8) | 0.292 |
| Currently married mothers–no. (%) | 134 (89.3) | 124 (83.8) | 0.217 |
| Maternal employment status–no. (%) | | | |
| Unemployed | 73 (48.7) | 75 (50.7) | 0.402 |
| Salaried or irregular working hours | 17 (11.3) | 23 (15.5) | |
| Self-employed | 60 (40.0) | 50 (33.8) | |
| Median income in 1000 KSH (IQR) | 10 (6–15) | 10 (6–15) | 0.598 |
| Delivered by c-section–no. (%) | 17 (11.4) | 26 (17.6) | 0.179 |
| Tobacco use during pregnancy–no. (%) | 1 | 0 | 1.000 |
| Alcohol use during pregnancy–no. (%) | 3 (2.0) | 6 (4.0) | 0.494 |
| Drug use during pregnancy–no. (%) | 5 (3.4) | 5 (3.4) | 0.999 |
| Timing of HIV diagnosis–no. (%) | | | |
| Before pregnancy | 115 (76.6) | 110 (74.3) | 0.841 |
| While pregnant for this child | 34 (22.7) | 37 (25.0) | |
| After delivery of this child | 1 (0.7) | 1 (0.7) | |
| Any IPT exposure during pregnancy–no. (%) | 77 (51.3) | 76 (51.4) | 1.00 |
| Ever diagnosed with tuberculosis–no. (%) | 15 (10.0) | 16 (10.8) | 0.967 |
| **Infant characteristics** | | | |
| Mean infant age in weeks (SD) | 6.6 (1.0) | 6.6 (1.2) | 0.882 |
| Female infants–no. (%) | 71 (47.3) | 71 (48.0) | 1.000 |
| Currently breastfeeding–no. (%) | 146 (97.3) | 147 (99.3) | 0.375 |
| Infant has BCG scar–no. (%) | 140 (93.3) | 136 (92.5) | 0.962 |
| Taking cotrimoxazole | 136 (90.7) | 132 (89.2) | 0.817 |
| Mean weight-for-age z-score | 0.2 (1.0) | 0.3 (1.0) | 0.503 |
| Underweight at enrollment | 3 (2) | 2 (1.3) | 1.000 |
| Mean height-for-age z-score | -0.2 (1.2) | -0.3 (1.3) | 0.176 |
| Stunted at enrollment | 9 (6.0) | 16 (10.9) | 0.148 |
| Mean weight-for-height z-score | 0.5 (1.6) | 0.9 (1.7) | **0.043** |
| Wasted at enrollment | 10 (6.7) | 5 (3.4) | 0.290 |

*IPT–isoniazid preventive therapy.

delivered via cesarean section. Only one mother smoked, 9 mothers drank alcohol, and 10 used drugs during pregnancy. All participating mothers were living with HIV, 23.8% (71/298) of whom were diagnosed with HIV during pregnancy, 73.2% (218/298) started ART before the pregnancy, 51.3% (153/298) had taken IPT during pregnancy, and 10.4% (31/298) had ever been diagnosed with TB.

The median age of the infants at enrollment was 6.3 weeks (6.0–6.6 weeks), and the median birthweight was 3.4 kg (IQR 3–3.7). All infants were of normal birth weight ($\geq$2.5 kg) due to eligibility criteria which excluded low birth weight infants. Of all infants, 47.6% (142/298) were females, 92.6% (276/298) had BCG scars, 90.0% (268/298) were on cotrimoxazole, and almost all, 99.0% (295/298), received ART for PMTCT after birth. Most (98.3% [293/298]) infants were breastfed. Infant mean WAZ (SD) was 0.2 (1.0), mean HAZ was -0.25 (1.2), and mean WHZ was 0.7 (1.6) at enrollment and similar between infants randomized to IPT or no-IPT. As reported in primary analyses, only 3.7% of infants in this study experienced gastroenteritis during the trial period [17].

## Changes in growth by ~14 and 24 months of age

There was a decline in growth z-scores, with average WAZ value of all infants decreasing by 0.33 ($\beta$ = -0.33 [95% CI: -0.48, -0.19]) and 0.58 ($\beta$ = -0.58 [95% CI: - 0.75, - 0.42]) z-scores by ~14 and 24 months of age, respectively. Similarly, the average HAZ of infants participating in this study decreased by 0.50 ($\beta$ = - 0.50 [95% CI: -0.66, -0.34]) and 1.74 z-scores ($\beta$ = - 1.74 [95% CI: - 1.99, - 1.52]) by ~14 and 24 months of age, respectively. WHZ decreased significantly by 0.40 ($\beta$ = -0.40[95% CI: -0.65, -0.16]) by ~14 months of age but the change was not significant ($\beta$ = 0.27 [95% CI: -0.02, 0.56]) by 24 months of age. There were no significant differences between the IPT vs. no-IPT arms in WAZ, HAZ, and WHZ at ~14 months and 24 months. Infants in the IPT arm had similar mean WAZ ($\beta$ = 0.22 [95% CI: -0.07, 0.50]), HAZ ($\beta$ = 0.08 [95% CI: -0.31, 0.47]), and WHZ ($\beta$ = 0.21 [95% CI: -0.19, 0.60]) at 24 months of age compared with children in the non-IPT arm. The observed mean change in growth is shown in Fig 1. At 24 months of age, 5.8% (13/225) were underweight, 47.9% (104/217) stunted, and 1.4% (3/215) wasted and did not differ significantly by arm.

## Effect of IPT on growth–WAZ, HAZ, and WHZ–at all follow-up timepoints

Multivariable models adjusted sex of infants, maternal education, age of mothers, timing of maternal HIV diagnosis, maternal IPT use during pregnancy, viral load, WAZ at baseline. Adjusted for all other factors, IPT use during infancy did not have a statistically significant effect on growth, either during the trial period up to ~14 months of age or through the observational follow-up until 24 months of age (p-value > 0.05). Infants in IPT arm had similar mean WAZ ($\beta$ to ~14 months of age 0.00 [95% CI: -0.19, 0.19] and $\beta$ to 24 months of age $\beta$ = 0.04 [95% CI: -0.14, 0.22]), HAZ ($\beta$ to ~14 months of age 0.16 [95% CI: -0.05, 0.37] and $\beta$ to 24 months of age $\beta$ = 0.13 [95% CI: -0.06, 0.33]), and WHZ ($\beta$ to ~14 months of age -0.14 [95% CI: -0.32, 0.05] and $\beta$ to 24 months of age $\beta$ = -0.08 [95% CI: -0.26, 0.10]) z-scores compared to infants in no-IPT arm. There was no statistically significant difference in rate of WAZ ($\beta$ to ~14 months of age 0.00 [95% CI: -0.01, 0.02] and $\beta$ to 24 months of age 0.01 [95% CI: 0.00, 0.02]), and HAZ ($\beta$ to ~14 months of age—0.01 [95% CI: -0.03, 0.01] and $\beta$ to 24 months of age -0.01 [95% CI: -0.02, 0.00]) change in a month between the randomized groups. However, infants in IPT arm had an increased rate of WHZ ($\beta$ to ~14 months of age 0.02 [95% CI: 0.00, 0.04] and $\beta$ to 24 months of age 0.02 [95% CI: 0.01, 0.04]) change in a month than infants in no-IPT arm (Table 2). The adjusted change in growth is also shown in Fig 1.

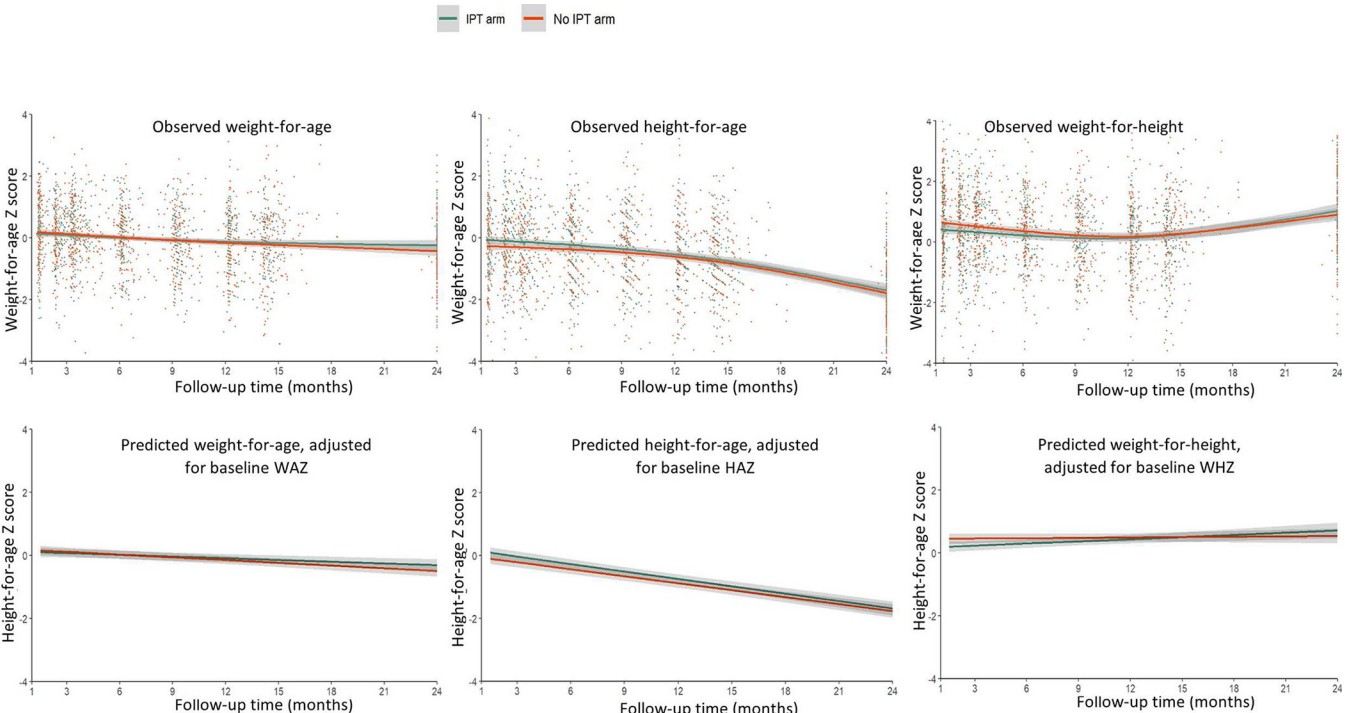

**Fig 1. Scatter plots of change in WAZ, HAZ, and WHZ, over time by the randomized arm.** (The Lowess curves represent a change in mean growth from randomization to 24 months of age. Bottom curves are adjusted change in WAZ, HAZ, and WHZ).

## Factors associated with growth change

Variables fitted in the model to identify factors associated with growth include infant IPT use, sex of infants and age, maternal educational status, timing of maternal HIV diagnosis (before vs. during pregnancy), any maternal IPT exposure during pregnancy, and viral load. Adjusted for all other variables, infants whose mothers completed secondary school had 0.26 WAZ (at ~14 months of age β = 0.26 [95% CI: 0.00, 0.52]) and HAZ (at 24 months of age β = 0.38 [95% CI: 0.00, 0.75]) higher than infants whose mothers completed primary school or less. Male infants had 0.53 HAZ (β = -0.53 [95% CI: -0.83, -0.23]) lower at ~14 months of age than female infants (S1 Table).

Adjusted for all factors, male infants had significantly lower WAZ (β to ~14 months of age -0.21 [95% CI: -0.04, 0.02] and β to 24 months of age -0.18 [95% CI: -0.36, -0.01]) and lower HAZ (β to ~14 months of age -0.34 [95% CI: -0.55, -0.13] and β to 24 months of age -0.28 [95% CI: -0.48, -0.09]) than female infants (Table 3).

## Discussion

In this secondary analysis of a randomized clinical trial, we found that IPT administered to HEU infants during the first year of life without known TB exposure did not significantly impact growth outcomes (WAZ, HAZ, and WHZ) during a 2-year follow-up. Infants in the IPT arm had similar mean WAZ, HAZ, and WHZ z-scores compared to infants in the no-IPT arm at the trial end to 14 months of age and through extended observational follow-up through 24 months of age. Despite the fact that isoniazid can cause side effects that could compromise caloric intake, including nausea, vomiting, and upset stomach [18], only 3.7% of the children in this study experienced gastroenteritis during trial period [16]. Our data suggest

**Table 2. Effect of IPT on growth–WAZ, HAZ, WHZ–from enrollment to ~14 and 24 months of age–intent-to-treat analysis–longitudinal and cross-sectional linear analyses.**

| Follow-up | Model | Randomization arm | WAZ[¶] at all follow-up timepoints (95% CI) | | HAZ[¥] at all follow-up timepoints (95% CI) | | WHZ at all follow-up timepoints (95% CI) | |
|---|---|---|---|---|---|---|---|---|
| | | | Coefficient (95% CI) | P-value | Coefficient (95% CI) | p-value | Coefficient (95% CI) | P-value |
| Longitudinal[§] up to ~14 months of age | Overall | IPT arm (ref: no-IPT) | 0.00 (-0.19, 0.19) | 0.985 | 0.16 (-0.05, 0.37) | 0.139 | -0.14 (-0.32, 0.05) | 0.139 |
| | Male | IPT arm (ref: no-IPT) | 0.02 (-0.27, 0.31) | 0.886 | 0.22 (-0.07, 0.52) | 0.141 | -0.18 (-0.45, 0.10) | 0.202 |
| | Female | IPT arm (ref: no-IPT) | -0.02 (-0.26, 0.21) | 0.852 | 0.09 (-0.19, 0.37) | 0.535 | -0.09 (-0.33, 0.14) | 0.435 |
| Longitudinal[§] up to ~24 months of age | Overall | IPT arm (ref: no-IPT) | 0.04 (-0.14, 0.22) | 0.657 | 0.13 (-0.06, 0.33) | 0.189 | -0.08 (-0.26, 0.10) | 0.399 |
| | Male | IPT arm (ref: no-IPT) | 0.07 (-0.20, 0.34) | 0.611 | 0.17 (-0.11, 0.46) | 0.231 | -0.09 (-0.36, 0.18) | 0.521 |
| | Female | IPT arm (ref: no-IPT) | 0.01 (-0.21, 0.24) | 0.927 | 0.09 (-0.17, 0.35) | 0.500 | -0.07 (-0.30, 0.17) | 0.578 |
| Cross-sectional[¤] at ~14 months of age | Overall | IPT arm (ref: no-IPT) | -0.01 (-0.28, 0.25) | 0.915 | 0.13 (-0.17, 0.42) | 0.392 | -0.11 (-0.42, 20) | 0.487 |
| | Male | IPT arm (ref: no-IPT) | 0.10 (-0.30, 0.50) | 0.610 | 0.32 (-0.09, 0.72) | 0.123 | -0.07 (-0.53, 0.38) | 0.751 |
| | Female | IPT arm (ref: no-IPT) | -0.14 (-0.48, 0.19) | 0.401 | -0.08 (-0.49, 0.33) | 0.697 | -0.15 (-0.58, 0.28) | 0.488 |
| Cross-sectional[¤] at 24 months of age | Overall | IPT arm (ref: no-IPT) | 0.19 (-0.08, 0.45) | 0.171 | 0.00 (-0.35, 0.35) | 0.995 | 0.25 (-0.13, 0.62) | 0.200 |
| | Male | IPT arm (ref: no-IPT) | 0.22 (-0.18, 0.62) | 0.275 | -0.11 (-0.60, 0.39) | 0.676 | 0.40 (-0.14, 0.94) | 0.148 |
| | Female | IPT arm (ref: no-IPT) | 0.15 (-0.20, 0.50) | 0.396 | 0.12 (-0.37, 0.60) | 0.636 | 0.08 (-0.43, 0.59) | 0.745 |

[¶]WAZ–weight-for-age z-score.

[¥]HAZ–height-for-age z-score.

WHZ–weight-for-height z-score.

*IPT–Isoniazid preventive therapy.

[§]Longitudinal–analysis included all data from 6th weeks to 14/24 months of age.

[¤]Cross-sectional–analysis is based on one-time data at 14/24 months of age.

that infant IPT did not impact growth, adding to safety data for IPT administered to infants [1–5]. The risk of exposure [8], progression, morbidity [21–24], and mortality [25–27] for Mtb infection is high among infants and even higher in those who are HEU children and CHIV. Approaches to prevent and treat TB infection in TB endemic areas in young children are important to decrease TB-related morbidity and mortality.

Considering the benefits of IPT in treating LTBI and preventing active TB disease in children [10–12] and people with HIV and without HIV, WHO recommends IPT to children with known TB exposure and children living with HIV (CHIV), older than one year of age [13]. However, most TB cases in children occur without a known contact [14] and therefore are missed by current WHO recommendations. The iTIPS trial did not find a significant benefit of IPT to prevent Mtb infection among HEU children [16, 17]; a post-hoc analysis demonstrated that with the observed ~10% cumulative incidence of Mtb infection (which was lower than anticipated at study design), there may not have been sufficient sample size to demonstrate benefit of IPT to prevent Mtb infection. It is plausible while IPT decreases progression to TB, it may not prevent Mtb infection. The analysis of the parent RCT demonstrated no

**Table 3. Factors associated with growth at ~14 and 24-month age–longitudinal.**

| Follow-up | Variable | WAZ¶ at all follow-up timepoints | | | HAZ¥ at all follow-up timepoints | | | WHZ at all follow-up timepoints | | |
|---|---|---|---|---|---|---|---|---|---|---|
| | | cCoefficient | aCoefficient* | P-value | cCoefficient | aCoefficient* | P-value | cCoefficient | aCoefficient* | P-value |
| Up to ~14 months of age | Infant in IPT-arm | -0.00 (-0.19, 0.19) | -0.04 (-0.22, 0.14) | 0.675 | 0.17 (-0.05, 0.38) | 0.15 (-0.06, 0.36) | 0.166 | -0.14 (-0.33, 0.04) | -0.18 (-0.37, 0.00) | 0.054 |
| | Male infants | -0.11 (-0.29, 0.08) | **-0.21 (-0.40, -0.02)** | **0.027** | -0.35 (-0.56, -0.14) | **-0.34 (-0.55, -0.13)** | **0.002** | 0.13 (-0.06, 0.31) | 0.07 (-0.12, 0.25) | 0.490 |
| | Secondary school or above | 0.13 (-0.06, 0.32) | 0.12 (-0.07, 0.30) | 0.210 | 0.07 (-0.15, 0.28) | -0.03 (-0.18, 0.24) | 0.771 | 0.13 (-0.05, 0.32) | 0.14 (-0.05, 0.32) | 0.141 |
| | Age of mothers | -0.01 (-0.03, 0.01) | -0.00 (-0.02, 0.01) | 0.646 | -0.02, -0.04, 0.00) | -0.02 (-0.04, 0.00) | 0.120 | 0.01 (-0.01, 0.03) | 0.01 (-0.01, 0.03) | 0.397 |
| | Tested HIV positive during pregnancy | 0.08 (-0.15, 0.30) | 0.06 (-0.17, 0.29) | 0.605 | 0.09 (-0.16, 0.34) | -0.03 (-0.30, 0.23) | 0.808 | 0.03 (-0.19, 0.25) | 0.11 (-0.12, 0.34) | 0.348 |
| | Pregnancy-IPT | 0.21 (0.02, 0.39) | 0.17 (-0.01, 0.36) | 0.062 | 0.12 (-0.10, 0.33) | 0.14 (-0.07, 0.35) | 0.182 | 0.14 (-0.04, 0.33) | 0.13 (-0.05, 0.32) | 0.166 |
| | Suppressed viral load (<40 copies/ml) | 0.08 (-0.17, 0.33) | 0.04 (-0.19, 0.28) | 0.706 | -0.01 (-0.28, 0.26) | 0.05 (-0.22, 0.32) | 0.730 | 0.10 (-0.14, 0.33) | 0.02 (-0.21, 0.26) | 0.846 |
| | WAZ at birth | 0.27 (0.18, 0.36) | 0.27(0.20, 0.38) | <0.001 | | | | 0.17 (0.09, 0.26) | **0.17 (0.08, 0.26)** | **<0.001** |
| Up to 24 months of age | Infant in IPT-arm | 0.04 (-0.14, 0.22) | -0.01 (-0.18, 0.17) | 0.945 | 0.14 (-0.06, 0.34) | 0.13 (-0.07, 0.33) | 0.215 | -0.08 (-0.26, 0.10) | -0.13 (-0.31, 0.05) | 0.143 |
| | Male infants | -0.08 (-0.26, 0.10) | -0.18 (-0.36, -0.01) | 0.041 | -0.29 (-0.49, 0.10) | **-0.28 (-0.48, -0.09)** | **0.005** | 0.11 (-0.07, 0.29) | 0.04 (-0.14, 0.22) | 0.693 |
| | Secondary school or above | 0.06 (-0.02, 0.33) | 0.15 (-0.02, 0.32) | 0.090 | 0.11 (-0.09, 0.31) | 0.09 (-0.11, 0.29) | 0.399 | 0.13 (-0.05, 0.31) | 0.13 (-0.05, 0.31) | 0.153 |
| | Age of mothers | -0.00 (-0.02, 0.01) | 0.00 (-0.02, 0.02) | 0.931 | -0.02 (-0.04, 0.00) | -0.02 (-0.04, 0.00) | 0.090 | 0.01 (-0.01, 0.03) | 0.01 (-0.00, 0.03) | 0.134 |
| | Tested HIV positive during pregnancy | 0.06 (-0.15, 0.27) | 0.08 (-0.14, 0.30) | 0.486 | 0.05 (-0.18, 0.28) | -0.04 (-0.29, 0.21) | 0.751 | 0.06 (-0.16, 0.27) | 0.14 (-0.08, 0.36) | 0.223 |
| | Pregnancy-IPT | 0.15 (-0.03, 0.32) | 0.12 (-0.06, 0.29) | 0.188 | 0.08 (-0.12, 0.27) | 0.11 (-0.09, 0.31) | 0.275 | 0.14 (-0.04, 0.32) | 0.12 (-0.06, 0.30) | 0.199 |
| | Suppressed viral load (<40 copies/ml) | 0.00 (-0.23, 0.23) | -0.04 (-0.26, 0.18) | 0.740 | -0.05 (-0.31, 0.20) | 0.01 (-0.26, 0.24) | 0.946 | 0.05 (-0.18, 0.28) | -0.01 (-0.24, 0.21) | 0.903 |
| | WAZ at birth | 0.24 (0.16, 0.33) | **0.26 (0.18, 0.35)** | **<0.001** | | | | 0.17 (0.08, 0.26) | **0.17 (0.08, 0.26)** | **<0.001** |

¶WAZ–weight-for-age.

¥HAZ–height-for-age.

WHZ–weight-for-height z-score.

IPT–Isoniazid preventive therapy.

ⁿcCoefficient—crude coefficient.

*aCoefficient–adjusted coefficient for all other variables (infant IPT-arm, sex of infants (ref: female infants), maternal education (ref: primary or less), age of mothers, timing of maternal HIV diagnosis (ref: tested positive before pregnancy), any maternal IPT exposure during pregnancy (ref: no use of IPT during pregnancy), viral load (ref: > = 40 copies/ml), and WAZ at baseline.

difference in adverse events between trial arms. Our new findings of comparable growth between arms provide additional evidence of safety applicable to pediatric IPT use in general.

In our study, WAZ and HAZ significantly declined after enrollment in all HEU cohort irrespective of trial arm. The mean HAZ of all infants significantly decreased by 1.74 Z-scores in 24 months. The proportion of stunting increased from 8.4% to 47.9% from baseline to 2 years of age. Similarly, studies conducted in Kenya [28] and Zimbabwe [29] found that 38.9% and 30% of HEU children are severely stunted, respectively. This implies that a significant number of HEU infants born with normal HAZ in this study became stunted within 24 months of their

birth. Stunting often indicates poor growth and development, and the fact that 'stunting can be irreversible after 1000 days of life makes [30] losing HAZ in the first 730 days a concern for HEU children.

A significant decline in HAZ was observed in male infants. This is in line with other findings [31]. Numerous epidemiological studies have consistently demonstrated higher morbidity and mortality rates among male children during early life [31–34]. The biological and social mechanisms driving higher neonatal mortality and morbidity in males and their greater HAZ decline compared to females remain undefined [34, 35], leaving the cause of this disparity unclear.

Our study had several strengths, with a strong study design–a randomized trial with over 90% retention. The weight and length of infants were measured twice, and the average values were used for analysis to reduce measurement error. This study also has limitations. Only normal birth weight and term infants were included in the parent trial, so results do not generalize to low-birthweight and preterm babies.

## Conclusion

In this secondary analysis of an RCT, IPT administered to healthy HEU infants ages 6–10 weeks over the first year of life without a known TB exposure did not significantly impact growth outcomes through the first 2 years of life. However, there was universally poor linear growth, especially in height, in this cohort of children with HIV exposure.

## Supporting information

**S1 Table. Factors associated with growth at ~14 a.**
(DOCX)

## Acknowledgments

Our sincere gratitude goes out to the study participants and their families.

## Author Contributions

**Conceptualization:** Sylvia M. LaCourse, John Kinuthia, Grace John-Stewart.

**Data curation:** Ashenafi Shumey Cherkos, Sylvia M. LaCourse, Jaclyn N. Escudero, Daniel Matemo, Grace John-Stewart.

**Formal analysis:** Ashenafi Shumey Cherkos, Sylvia M. LaCourse, Daniel A. Enquobahrie, Grace John-Stewart.

**Funding acquisition:** Sylvia M. LaCourse, John Kinuthia, Grace John-Stewart.

**Investigation:** Sylvia M. LaCourse, Jaclyn N. Escudero, Jerphason Mecha, Daniel Matemo, John Kinuthia, Grace John-Stewart.

**Methodology:** Ashenafi Shumey Cherkos, Sylvia M. LaCourse, Daniel A. Enquobahrie, Jaclyn N. Escudero, Jerphason Mecha, John Kinuthia, Grace John-Stewart.

**Project administration:** Sylvia M. LaCourse, Jaclyn N. Escudero, Jerphason Mecha, Daniel Matemo, John Kinuthia, Grace John-Stewart.

**Resources:** Sylvia M. LaCourse, Jaclyn N. Escudero, John Kinuthia, Grace John-Stewart.

**Software:** Sylvia M. LaCourse, Jaclyn N. Escudero, Grace John-Stewart.

**Supervision:** Sylvia M. LaCourse, Jaclyn N. Escudero, Jerphason Mecha, Daniel Matemo, John Kinuthia, Grace John-Stewart.

**Validation:** Jaclyn N. Escudero, Jerphason Mecha, Daniel Matemo, John Kinuthia, Grace John-Stewart.

**Visualization:** Ashenafi Shumey Cherkos, Daniel A. Enquobahrie, Jaclyn N. Escudero.

**Writing – original draft:** Ashenafi Shumey Cherkos, Daniel A. Enquobahrie.

**Writing – review & editing:** Ashenafi Shumey Cherkos, Daniel A. Enquobahrie, Jaclyn N. Escudero, Jerphason Mecha, Daniel Matemo, John Kinuthia, Sarah J. Iribarren.

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
