## [Decision Letter · Decision Letter 0]

12 Jan 2024

PONE-D-23-31679Isoniazid preventive therapy during infancy does not adversely effect growth among HIV-exposed uninfected children: secondary analysis of data from a randomized controlled trialPLOS ONE

Dear Dr. Cherkos,

Thank you for submitting your manuscript to PLOS ONE. After careful consideration, we feel that it has merit but does not fully meet PLOS ONE’s publication criteria as it currently stands. Therefore, we invite you to submit a revised version of the manuscript that addresses the points raised during the review process.

We look forward to receiving your revised manuscript.

Kind regards,

Tinei Shamu

Academic Editor

PLOS ONE

Journal Requirements:

   "This work was supported by the Thrasher Research Fund, National Institute of Allergy and Infectious Diseases (NIAID), Fulbright program awarded to the Northern Pacific Global Health Fellows Program by the Fogarty International Center of the National Institutes of Health (NIH/Fogarty), and National Center for Advancing Translational Sciences at National Institutes of Health (NIH) (Thrasher to GJ-S, NIH/NIAID K23AI120793 to SML, NIH/NIAID 2K24AI137310 to TRH, NIH/Fogarty R25TW009345 to AJW and NIH UL1TR000423 for REDCap). ASC had a diversity supplement grant from NIH (NIH/NIAID 1R01AI142647)."  

6. We note that you have indicated that there are restrictions to data sharing for this study. PLOS only allows data to be available upon request if there are legal or ethical restrictions on sharing data publicly. For more information on unacceptable data access restrictions, please see http://journals.plos.org/plosone/s/data-availability#loc-unacceptable-data-access-restrictions. 

Reviewers' comments:

Reviewer's Responses to Questions

**Comments to the Author**

1. Is the manuscript technically sound, and do the data support the conclusions?

Reviewer #1: Yes

Reviewer #2: Yes

2. Has the statistical analysis been performed appropriately and rigorously? 

Reviewer #1: I Don't Know

Reviewer #2: Yes

3. Have the authors made all data underlying the findings in their manuscript fully available?

Reviewer #1: Yes

Reviewer #2: No

4. Is the manuscript presented in an intelligible fashion and written in standard English?

Reviewer #1: Yes

Reviewer #2: Yes

5. Review Comments to the Author

Reviewer #1: General

This is a clear well written and informative paper. It is easy to follow and understand and the authors clearly demonstrate why the study was carried out, its importance, how the study was carried out, the findings/results and how the results were interpreted. The study question is answered appropriately.

In this study the authors do a secondary data analysis on data collected during a randomized control trial and compare the growth outcomes of HIV exposed uninfected (HEU) infants who received isoniazid preventive therapy (IPT) for the first year of their life with those who did not receive any IPT. They found that IPT did nit significantly affect the growth outcomes of infants in the first two years of their life.

Abstract

The authors summarise the main research question and key findings well

Introduction

The authors give us the importance of the study - and they explain about the added advantage of leveraging on the RTC design to be able to investigate the effect of IPT on growth of HEU infants without introducing the confounding brought about by other factors such as known exposure to TB.

Figures and tables

The bottom right Lowess curve is mislabelled. it is labelled "predicted height-for-age, adjusted for baseline HAZ" instead of "predicted weight-for-height, adjusted for baseline WHZ".

Methods

The parent trial and design and intervention is explained well. Ethical protocols were observed.

However, the specific tools that were used to measure the infant’s weight and height are not described.

Statistical methods – The authors could perhaps mention why they use LMEMs with autoregression correlation structure and what it entails.

The authors explain how they dealt with missingness which is good.

Results, discussion, conclusion

Generally, the results support the conclusion. Limitations are discussed appropriately. Previous studies and supporting data is shared .

However, the information in line 245 – 248 somewhat seems misplaced and confusing. It does not quite make sense how the mortality of male infants is explained by the value placed on females in the society and how this is all related to the observed finding of declining HAZ in male infants.

Reviewer #2: Isoniazid preventive therapy during infancy does not adversely effect growth among HIV-exposed uninfected children: secondary analysis of data from a randomized controlled trial

Summary

This is a secondary analysis of data from a randomized controlled trial assessing growth among HIV exposed uninfected children in Western Kenya with and without INH prophylaxis. There were no differences in growth between children receiving INH and those who did not. Significant stunting was observed in both arms by 24 months of age.

Comments

General: This is an important study showing the safety of INH prophylaxis initiated in infants from 6-10 weeks of age. While the benefits of this prophylaxis are still under study as demonstrated in the primary study and its limitations, the current study resolves the question of safety with specific focus on growth indicators.

Specific comments:

Title - Consider replacing “effect” with “affect” as “adversely effect” seems to imply causing something that was not expected or desired whereas “adversely affect” implies altering an expected process to a less desirable or undesirable state.

Line 3: Consider rephrasing to “Without treatment, children with latent Mtb infection (LTBI) have about…” as it is not true that Mtb infection is also called latent TB infection.

Line 19-20: there is a missing closing parenthesis.

Infant growth characterization: Consider adding more detail on the tools and methods used for measurement of height and weight as these are the main measurements for this study. Did all the clinics use similar tools?

Are these clinics in urban or rural areas and are they public health clinics?

Line 121: Rephrase to “All participating mothers were living with HIV…”

Table 2: Consider standardizing placement of symbols for WAZ, HAZ, and WHZ

Line 228: There’s a missing period.

Lines 237 – 242: Is this proportion of stunting consistent with other studies of HEU children? Consider citations https://www.sciencedirect.com/science/article/pii/S2405844019357330, https://www.ncbi.nlm.nih.gov/pmc/articles/PMC7868486/, https://onlinelibrary.wiley.com/doi/full/10.1002/fsn3.2509

6. PLOS authors have the option to publish the peer review history of their article (what does this mean?). If published, this will include your full peer review and any attached files.

Reviewer #1: **Yes: **Linda Mandikiyana Chirimuta

Reviewer #2: No

---

## [Author Response · Author response to Decision Letter 0]

24 Mar 2024

PONE-D-23-31679

Isoniazid preventive therapy during infancy does not adversely effect growth among HIV-exposed uninfected children: secondary analysis of data from a randomized controlled trial

PLOS ONE

Journal Requirements:

Response: Edited accordingly.

Response: Due to the confidentiality agreement the participants signed, the data cannot be deposited, and de-identified data will be provided upon request.

Response: We have removed funding-related text from the manuscript. 

4. We note that the grant information you provided in the 'Funding Information' and 'Financial Disclosure' sections do not match. When you resubmit, please ensure that you provide the correct grant numbers for the awards you received for your study in the 'Funding Information' section.

Response: We have corrected grant numbers as noted. 

5. Thank you for stating the following financial disclosure: "This work was supported by the Thrasher Research Fund, National Institute of Allergy and Infectious Diseases (NIAID), Fulbright program awarded to the Northern Pacific Global Health Fellows Program by the Fogarty International Center of the National Institutes of Health (NIH/Fogarty), and National Center for Advancing Translational Sciences at National Institutes of Health (NIH) (Thrasher to GJ-S, NIH/NIAID K23AI120793 to SML, NIH/NIAID 2K24AI137310 to TRH, NIH/Fogarty R25TW009345 to AJW and NIH UL1TR000423 for REDCap). ASC had a diversity supplement grant from NIH (NIH/NIAID 1R01AI142647)." Please state what role the funders took in the study. 

Response: We would like to change the above content by “This work was supported by the Thrasher Research Fund, National Institute of Allergy and Infectious Diseases (NIAID), and National Center for Advancing Translational Sciences at National Institutes of Health (NIH) (Thrasher to GJ-S, NIH/NIAID K23AI120793 to SML). ASC had a diversity supplement grant from NIH (NIH/NIAID 1R01AI142647). The funders had no role in study design, data collection and analysis, decision to publish, or preparation of the manuscript. The funders had no role in study design, data collection and analysis, decision to publish, or preparation of the manuscript."

6. We note that you have indicated that there are restrictions to data sharing for this study. PLOS only allows data to be available upon request if there are legal or ethical restrictions on sharing data publicly. For more information on unacceptable data access restrictions, please see http://journals.plos.org/plosone/s/data-availability#loc-unacceptable-data-access-restrictions. 

Response: We would confirm with the Kenyatta National Hospital-University of Nairobi ERC prior to provision of de-identified data. Interested individuals should contact, 

STEPHANIE EDLUND-CHO, MSW

Program Operations Specialist | Global WACh

Department of Global Health 

Hans Rosling Center Box 351620

3980 15th Ave NE, Seattle, WA 98195

office: 206.685.6809

globalwach.org

https://journals.plos.org/plosone/s/recommended-repositories. You also have the option of uploading the data as Supporting Information files, but we would recommend depositing data directly to a data repository if possible. We will update your Data Availability statement on your behalf to reflect the information you provide.

Response: see 6a.

Response: added “Supplementary document: S1 Table 1: Factors associated with growth at ~14 and 24-month age"

Reviewers' comments:

Reviewer's Responses to Questions

Comments to the Author

1. Is the manuscript technically sound, and do the data support the conclusions?

Reviewer #1: Yes

Reviewer #2: Yes

2. Has the statistical analysis been performed appropriately and rigorously?

Reviewer #1: I Don't Know

Reviewer #2: Yes

3. Have the authors made all data underlying the findings in their manuscript fully available?

Reviewer #1: Yes

Reviewer #2: No

4. Is the manuscript presented in an intelligible fashion and written in standard English?

Reviewer #1: Yes

Reviewer #2: Yes

5. Review Comments to the Author

Reviewer #1: General

This is a clear well written and informative paper. It is easy to follow and understand and the authors clearly demonstrate why the study was carried out, its importance, how the study was carried out, the findings/results and how the results were interpreted. The study question is answered appropriately.

In this study the authors do a secondary data analysis on data collected during a randomized control trial and compare the growth outcomes of HIV exposed uninfected (HEU) infants who received isoniazid preventive therapy (IPT) for the first year of their life with those who did not receive any IPT. They found that IPT did nit significantly affect the growth outcomes of infants in the first two years of their life.

Abstract

The authors summarise the main research question and key findings well

Introduction

The authors give us the importance of the study - and they explain about the added advantage of leveraging on the RTC design to be able to investigate the effect of IPT on growth of HEU infants without introducing the confounding brought about by other factors such as known exposure to TB.

Figures and tables

The bottom right Lowess curve is mislabelled. it is labelled "predicted height-for-age, adjusted for baseline HAZ" instead of "predicted weight-for-height, adjusted for baseline WHZ".

Response: Thanks for noticing this error. We have corrected the mislabeled noted. 

Methods

The parent trial and design and intervention is explained well. Ethical protocols were observed.

However, the specific tools that were used to measure the infant's weight and height are not described.

Statistical methods – The authors could perhaps mention why they use LMEMs with autoregression correlation structure and what it entails.

The authors explain how they dealt with missingness which is good.

Response: We used conventional growth measurement tools – a scale for weight and measuring tape for height. All data collectors were trained by the CDC-Kenya team on how to measure an infant's weight and height.

Results, discussion, conclusion

Generally, the results support the conclusion. Limitations are discussed appropriately. Previous studies and supporting data is shared.

However, the information in line 245 – 248 somewhat seems misplaced and confusing. It does not quite make sense how the mortality of male infants is explained by the value placed on females in the society and how this is all related to the observed finding of declining HAZ in male infants.

Response: edited and added “The biological and social mechanisms driving higher neonatal mortality and morbidity in males and their greater HAZ decline compared to females remain undefined (34,35), leaving the cause of this disparity unclear.”

Reviewer #2: Isoniazid preventive therapy during infancy does not adversely effect growth among HIV-exposed uninfected children: secondary analysis of data from a randomized controlled trial

Summary

This is a secondary analysis of data from a randomized controlled trial assessing growth among HIV exposed uninfected children in Western Kenya with and without INH prophylaxis. There were no differences in growth between children receiving INH and those who did not. Significant stunting was observed in both arms by 24 months of age.

Comments

General: This is an important study showing the safety of INH prophylaxis initiated in infants from 6-10 weeks of age. While the benefits of this prophylaxis are still under study as demonstrated in the primary study and its limitations, the current study resolves the question of safety with specific focus on growth indicators.

Specific comments:

Title - Consider replacing "effect" with "affect" as "adversely effect" seems to imply causing something that was not expected or desired whereas "adversely affect" implies altering an expected process to a less desirable or undesirable state.

Response: Replaced "effect" with "affect."

Line 3: Consider rephrasing to "Without treatment, children with latent Mtb infection (LTBI) have about…" as it is not true that Mtb infection is also called latent TB infection.

Response: Thanks for the comment: We edited it accordingly: "Without treatment, children with latent Mtb infection, also called latent TB infection (LTBI)"

Line 19-20: there is a missing closing parenthesis.

Response: Thanks for noting this. The closing parenthesis is added. 

Infant growth characterization: Consider adding more detail on the tools and methods used for measurement of height and weight as these are the main measurements for this study. Did all the clinics use similar tools?

Are these clinics in urban or rural areas and are they public health clinics?

Response: All data collectors were trained by the CDC-Kenya team on how to measure an infant's weight and height. Under the participants and study period subheading, we added: "All institutions in the study were public maternal and child health (MCH) clinics embedded in Kenya's Ministry of Health (MOH)."

Line 121: Rephrase to "All participating mothers were living with HIV…"

 Response: Thanks for the comment: we changed it accordingly.

Table 2: Consider standardizing placement of symbols for WAZ, HAZ, and WHZ

Response: Thank you for the comment: placement standardized.

Line 228: There's a missing period.

Response: Thank you for the comment. Period added. 

Lines 237 – 242: Is this proportion of stunting consistent with other studies of HEU children? Consider citations https://www.sciencedirect.com/science/article/pii/S2405844019357330, https://www.ncbi.nlm.nih.gov/pmc/articles/PMC7868486/, https://onlinelibrary.wiley.com/doi/full/10.1002/fsn3.2509

Response: We cited two of the recommended papers that have follow-up periods similar to ours. We added, "Similarly, studies conducted in Kenya (28) and Zimbabwe (29) found that 38.9% and 30% of HEU children are severely stunted, respectively."

6. PLOS authors have the option to publish the peer review history of their article (what does this mean?). If published, this will include your full peer review and any attached files.

Do you want your identity to be public for this peer review? For information about this choice, including consent withdrawal, please see our Privacy Policy.

Reviewer #1: Yes: Linda Mandikiyana Chirimuta

Reviewer #2: No

---

## [Decision Letter · Decision Letter 1]

9 Apr 2024

Isoniazid preventive therapy during infancy does not adversely affect growth among HIV-exposed uninfected children: secondary analysis of data from a randomized controlled trial

PONE-D-23-31679R1

Dear Dr. Cherkos,

We’re pleased to inform you that your manuscript has been judged scientifically suitable for publication and will be formally accepted for publication once it meets all outstanding technical requirements.

Kind regards,

Steve Graham

Stephen Michael Graham, FRACP, PhD

Academic Editor

PLOS ONE

Additional Editor Comments (optional):

Reviewers' comments:

Reviewer's Responses to Questions

**Comments to the Author**

1. If the authors have adequately addressed your comments raised in a previous round of review and you feel that this manuscript is now acceptable for publication, you may indicate that here to bypass the “Comments to the Author” section, enter your conflict of interest statement in the “Confidential to Editor” section, and submit your "Accept" recommendation.

Reviewer #1: All comments have been addressed

2. Is the manuscript technically sound, and do the data support the conclusions?

Reviewer #1: Yes

3. Has the statistical analysis been performed appropriately and rigorously? 

Reviewer #1: I Don't Know

4. Have the authors made all data underlying the findings in their manuscript fully available?

Reviewer #1: No

5. Is the manuscript presented in an intelligible fashion and written in standard English?

Reviewer #1: Yes

6. Review Comments to the Author

Reviewer #1: (No Response)

7. PLOS authors have the option to publish the peer review history of their article (what does this mean?). If published, this will include your full peer review and any attached files.

Reviewer #1: **Yes: **Linda Mandikiyana Chirimuta

---

## [Editor Report · Acceptance letter]

8 Aug 2024

PONE-D-23-31679R1 

PLOS ONE

Dear Dr. John-Stewart, 

I'm pleased to inform you that your manuscript has been deemed suitable for publication in PLOS ONE. Congratulations! Your manuscript is now being handed over to our production team.

Kind regards, 

on behalf of

Dr. Stephen Michael Graham 

Academic Editor

PLOS ONE